# Quartz: Randomized Dual Coordinate Ascent with Arbitrary Sampling

**Zheng Qu**
Department of Mathematics
The University of Hong Kong
Hong Kong
zhengqu@maths.hku.hk

**Peter Richtárik**
School of Mathematics
The University of Edinburgh
EH9 3FD, United Kingdom
peter.richtarik@ed.ac.uk

**Tong Zhang**
Department of Statistics
Rutgers University
Piscataway, NJ, 08854
tzhang@stat.rutgers.edu

## Abstract

We study the problem of minimizing the average of a large number of smooth convex functions penalized with a strongly convex regularizer. We propose and analyze a novel primal-dual method (Quartz) which at every iteration samples and updates a random subset of the dual variables, chosen according to an *arbitrary distribution*. In contrast to typical analysis, we directly bound the decrease of the primal-dual error (in expectation), without the need to first analyze the dual error. Depending on the choice of the sampling, we obtain efficient serial and mini-batch variants of the method. In the serial case, our bounds match the best known bounds for SDCA (both with uniform and importance sampling). With standard mini-batching, our bounds predict initial data-independent speedup as well as *additional data-driven speedup* which depends on spectral and sparsity properties of the data.

**Keywords:** empirical risk minimization, dual coordinate ascent, arbitrary sampling, data-driven speedup.

## 1 Introduction

In this paper we consider a primal-dual pair of structured convex optimization problems which has in several variants of varying degrees of generality attracted a lot of attention in the past few years in the machine learning and optimization communities [4, 22, 20, 23, 21, 27].

Let $A_1, \ldots, A_n$ be a collection of $d$-by-$m$ real matrices and $\phi_1, \ldots, \phi_n$ be $1/\gamma$-smooth convex functions from $\mathbb{R}^m$ to $\mathbb{R}$, where $\gamma > 0$. Further, let $g : \mathbb{R}^d \to \mathbb{R} \cup \{+\infty\}$ be a 1-strongly convex function and $\lambda > 0$ a regularization parameter. We are interested in solving the following *primal* problem:

$$\min_{w=(w_1,\ldots,w_d)\in\mathbb{R}^d} \left[ P(w) \stackrel{\text{def}}{=} \tfrac{1}{n} \sum_{i=1}^n \phi_i(A_i^\top w) + \lambda g(w) \right]. \tag{1}$$

In the machine learning context, matrices $\{A_i\}$ are interpreted as examples/samples, $w$ is a (linear) predictor, function $\phi_i$ is the loss incurred by the predictor on example $A_i$, $g$ is a regularizer, $\lambda$ is a regularization parameter and (1) is the *regularized empirical risk minimization* problem. In

this paper we are especially interested in problems where $n$ is very big (millions, billions), and much larger than $d$. This is often the case in *big data* applications. Stochastic Gradient Descent (SGD) [18, 11, 25] was designed for solving this type of large-scale optimization problems. In each iteration SGD computes the gradient of one single randomly chosen function $\phi_i$ and approximates the gradient using this unbiased but noisy estimation. Because of the variance of the stochastic estimation, SGD has slow convergence rate $O(1/\epsilon)$. Recently, many methods achieving fast (linear) convergence rate $O(\log(1/\epsilon))$ have been proposed, including SAG [19], SVRG [6], S2GD [8], SAGA [1], mS2GD [7] and MISO [10], all using different techniques to reduce the variance.

Another approach, such as Stochastic Dual Coordinate Ascent (SDCA) [22], solves (1) by considering its dual problem that is defined as follows. For each $i$, let $\phi_i^* : \mathbb{R}^m \to \mathbb{R}$ be the convex conjugate of $\phi_i$, namely, $\phi_i^*(u) = \max_{s \in \mathbb{R}^m} s^\top u - \phi_i(s)$ and similarly let $g^* : \mathbb{R}^d \to \mathbb{R}$ be the convex conjugate of $g$. The *dual problem* of (1) is defined as:

$$\max_{\alpha=(\alpha_1,\ldots,\alpha_n)\in\mathbb{R}^N=\mathbb{R}^{nm}} \left[ D(\alpha) \stackrel{\text{def}}{=} -f(\alpha) - \psi(\alpha) \right], \tag{2}$$

where $\alpha = (\alpha_1, \ldots, \alpha_n) \in \mathbb{R}^N = \mathbb{R}^{nm}$ is obtained by stacking dual variables (blocks) $\alpha_i \in \mathbb{R}^m$, $i = 1, \ldots, n$, on top of each other and functions $f$ and $\psi$ are defined by

$$f(\alpha) \stackrel{\text{def}}{=} \lambda g^* \left( \tfrac{1}{\lambda n} \sum_{i=1}^n A_i \alpha_i \right); \qquad \psi(\alpha) \stackrel{\text{def}}{=} \tfrac{1}{n} \sum_{i=1}^n \phi_i^*(-\alpha_i). \tag{3}$$

SDCA [22] and its proximal extension Prox-SDCA [20] first solve the dual problem (2) by updating uniformly at random one dual variable at each round and then recover the primal solution by setting $w = \nabla g^*(\alpha)$. Let $L_i = \lambda_{\max}(A_i^\top A_i)$. It is known that if we run SDCA for at least

$$O \left( \left( n + \tfrac{\max_i L_i}{\lambda\gamma} \right) \log \left( \left( n + \tfrac{\max_i L_i}{\lambda\gamma} \right) \tfrac{1}{\epsilon} \right) \right)$$

iterations, then SDCA finds a pair $(w, \alpha)$ such that $\mathbb{E}[P(w) - D(\alpha)] \leq \epsilon$. By applying accelerated randomized coordinate descent on the dual problem, APCG [9] needs at most $\tilde{O}(n + \sqrt{\frac{\max_i L_i}{\lambda\gamma}})$ number of iterations to get $\epsilon$-accuracy. ASDCA [21] and SPDC [26] are also accelerated and randomized primal-dual methods. Moreover, they can update a mini-batch of dual variables in each round.

We propose a new algorithm (Algorithm 1), which we call Quartz, for simultaneously solving the primal (1) and dual (2) problems. On the dual side, at each iteration our method selects and updates a *random subset (sampling)* $\hat{S} \subseteq \{1, \ldots, n\}$ of the dual variables/blocks. We assume that these sets are i.i.d. throughout the iterations. However, *we do not impose any additional assumptions on the distribution of $\hat{S}$* apart from the necessary requirement that each block $i$ needs to be chosen with a positive probability: $p_i \stackrel{\text{def}}{=} \mathbb{P}(i \in \hat{S}) > 0$. Quartz is the first SDCA-like method analyzed for an *arbitrary sampling*. The dual updates are then used to perform an update to the primal variable $w$ and the process is repeated. Our primal updates are different (less aggressive) from those used in SDCA [22] and Prox-SDCA [20], thanks to which the decrease in the primal-dual error can be bounded directly without first establishing the dual convergence as in [20], [23] and [9]. Our analysis is novel and *directly primal-dual* in nature. As a result, our proof is more direct, and the logarithmic term in our bound has a simpler form.

**Main result.** We prove that starting from an initial pair $(w^0, \alpha^0)$, Quartz finds a pair $(w, \alpha)$ for which $P(w) - D(\alpha) \leq \epsilon$ (in expectation) in at most

$$\max_i \left( \tfrac{1}{p_i} + \tfrac{v_i}{p_i \lambda \gamma n} \right) \log \left( \tfrac{P(w^0) - D(\alpha^0)}{\epsilon} \right) \tag{4}$$

iterations. The parameters $v_1, \ldots, v_n$ are assumed to satisfy the following ESO (expected separable overapproximation) inequality:

$$\mathbb{E}_{\hat{S}} \left[ \left\| \sum_{i \in \hat{S}} A_i h_i \right\|^2 \right] \leq \sum_{i=1}^n p_i v_i \|h_i\|^2, \tag{5}$$

where $\|\cdot\|$ denotes the standard Euclidean norm. Moreover, the parameters $v_1, \ldots, v_n$ are needed to run the method (they determine stepsizes), and hence it is critical that they can be cheaply computed

before the method starts. We wish to point out that (5) always holds for *some* parameters $\{v_i\}$. Indeed, the left hand side is a quadratic function of $h$ and hence the inequality holds for large-enough $v_i$. Having said that, the size of these parameters directly influences the complexity, and hence one would want to obtain as tight bounds as possible. As we will show, for many samplings of interest small enough parameter $v$ can be obtained in time required to read the data $\{A_i\}$. In particular, if the data matrix $A = (A_1, \ldots, A_n)$ is sufficiently sparse, our iteration complexity result (4) specialized to the case of standard mini-batching can be better than that of accelerated methods such as ASDCA [21] and SPDC [26] even when the condition number $\max_i L_i/\lambda\gamma$ is larger than $n$, see Proposition 4 and Figure 2.

As described above, Quartz uses an *arbitrary sampling* for picking the dual variables to be updated in each iteration. To the best of our knowledge, only two papers exist in the literature where a stochastic method using an arbitrary sampling was analyzed: NSync [16] for unconstrained minimization of a strongly convex function and ALPHA [15] for composite minimization of non-strongly convex function. Assumption (5) was for the first time introduced in [16]. However, NSync is not a primal-dual method. Besides NSync, the closest works to ours in terms of the generality of the sampling are PCDM [17], SPCDM [3] and APPROX [2]. All these are randomized coordinate descent methods, and all were analyzed for arbitrary *uniform* samplings (i.e., samplings satisfying $\mathbb{P}(i \in \hat{S}) = \mathbb{P}(i' \in \hat{S})$ for all $i, i' \in \{1, \ldots, n\}$). Again, none of these methods were analyzed in a primal-dual framework.

In Section 2 we describe the algorithm, show that it admits a natural interpretation in terms of Fenchel duality and discuss the flexibility of Quartz. We then proceed to Section 3 where we state the main result, specialize it to the samplings discussed in Section 2, and give detailed comparison of our results with existing results for related primal-dual stochastic methods in the literature. In Section 4 we demonstrate how Quartz compares to other related methods through numerical experiments.

## 2    The Quartz Algorithm

Throughout the paper we consider the standard Euclidean norm, denoted by $\| \cdot \|$. A function $\phi : \mathbb{R}^m \to \mathbb{R}$ is $(1/\gamma)$-smooth if it is differentiable and has Lipschitz continuous gradient with Lispchitz constant $1/\gamma$: $\|\nabla\phi(x) - \nabla\phi(y)\| \leq \frac{1}{\gamma}\|x - y\|$, for all $x, y \in \mathbb{R}^m$. A function $g : \mathbb{R}^d \to \mathbb{R} \cup \{+\infty\}$ is 1-strongly convex if $g(w) \geq g(w') + \langle\nabla g(w'), w - w'\rangle + \frac{1}{2}\|w - w'\|^2$ for all $w, w' \in \mathrm{dom}(g)$, where $\mathrm{dom}(g)$ denotes the domain of $g$ and $\nabla g(w')$ is a subgradient of $g$ at $w'$.

The most important parameter of Quartz is a random sampling $\hat{S}$, which is a random subset of $[n] = \{1, 2, \ldots, n\}$. The only assumption we make on the sampling $\hat{S}$ in this paper is the following:

**Assumption 1 (Proper sampling)** $\hat{S}$ *is a proper sampling; that is,*

$$p_i \stackrel{def}{=} \mathbb{P}(i \in \hat{S}) > 0, \qquad i \in [n]. \tag{6}$$

This assumption guarantees that each block (dual variable) has a chance to get updated by the method. Prior to running the algorithm, we compute positive constants $v_1, \ldots, v_n$ satisfying (5) to define the stepsize parameter $\theta$ used throughout in the algorithm:

$$\theta = \min_i \frac{p_i\lambda\gamma n}{v_i + \lambda\gamma n}. \tag{7}$$

Note from (5) that $\theta$ depends on both the data matrix $A$ and the sampling $\hat{S}$. We shall show how to compute in less than two passes over the data the parameter $v$ satisfying (5) for some examples of sampling in Section 2.2.

### 2.1    Interpretation of Quartz through Fenchel duality

---

**Algorithm 1** Quartz

---

**Parameters**: proper random sampling $\hat{S}$ and a positive vector $v \in \mathbb{R}^n$
**Initialization**: $\alpha^0 \in \mathbb{R}^N$; $w^0 \in \mathbb{R}^d$; $p_i = \mathbb{P}(i \in \hat{S})$; $\theta = \min_i \frac{p_i \lambda \gamma n}{v_i + \lambda \gamma n}$; $\bar{\alpha}^0 = \frac{1}{\lambda n} \sum_{i=1}^{n} A_i \alpha_i^0$

**for** $t \geq 1$ **do**
$\quad w^t = (1 - \theta) w^{t-1} + \theta \nabla g^*(\bar{\alpha}^{t-1})$
$\quad \alpha^t = \alpha^{t-1}$
$\quad$ Generate a random set $S_t \subseteq [n]$, following the distribution of $\hat{S}$
$\quad$ **for** $i \in S_t$ **do**
$\quad\quad \alpha_i^t = (1 - \theta p_i^{-1}) \alpha_i^{t-1} - \theta p_i^{-1} \nabla \phi_i(A_i^\top w^t)$
$\quad$ **end for**
$\quad \bar{\alpha}^t = \bar{\alpha}^{t-1} + (\lambda n)^{-1} \sum_{i \in S_t} A_i(\alpha_i^t - \alpha_i^{t-1})$
**end for**
**Output:** $w^t, \alpha^t$

---

Quartz (Algorithm 1) has a natural interpretation in terms of Fenchel duality. Let $(w, \alpha) \in \mathbb{R}^d \times \mathbb{R}^N$ and define $\bar{\alpha} = \frac{1}{\lambda n} \sum_{i=1}^{n} A_i \alpha_i$. The duality gap for the pair $(w, \alpha)$ can be decomposed as:

$$
\begin{aligned}
P(w) - D(\alpha) \quad &\overset{(1)+(2)}{=} \quad \lambda \left( g(w) + g^*(\bar{\alpha}) \right) + \frac{1}{n} \sum_{i=1}^{n} \phi_i(A_i^\top w) + \phi_i^*(-\alpha_i) \\
&= \quad \lambda \underbrace{\left( g(w) + g^*(\bar{\alpha}) - \langle w, \bar{\alpha} \rangle \right)}_{GAP_g(w,\alpha)} + \frac{1}{n} \sum_{i=1}^{n} \underbrace{\phi_i(A_i^\top w) + \phi_i^*(-\alpha_i) + \langle A_i^\top w, \alpha_i \rangle}_{GAP_{\phi_i}(w,\alpha_i)}.
\end{aligned}
$$

By Fenchel-Young inequality, $GAP_g(w, \alpha) \geq 0$ and $GAP_{\phi_i}(w, \alpha_i) \geq 0$ for all $i$, which proves weak duality for the problems (1) and (2), i.e., $P(w) \geq D(\alpha)$. The pair $(w, \alpha)$ is optimal when both $GAP_g$ and $GAP_{\phi_i}$ for all $i$ are zero. It is known that this happens precisely when the following optimality conditions hold:

$$w = \nabla g^*(\bar{\alpha}) \tag{8}$$

$$\alpha_i = -\nabla \phi_i(A_i^\top w), \quad i \in [n]. \tag{9}$$

We will now interpret the primal and dual steps of Quartz in terms of the above discussion. It is easy to see that Algorithm 1 updates the primal and dual variables as follows:

$$w^t = (1 - \theta) w^{t-1} + \theta \nabla g^*(\bar{\alpha}^{t-1}) \tag{10}$$

$$
\alpha_i^t = \left\{
\begin{array}{ll}
\left( 1 - \theta p_i^{-1} \right) \alpha_i^{t-1} + \theta p_i^{-1} \left( -\nabla \phi_i(A_i^\top w^t) \right) & , \; i \in S_t \\
\alpha_i^{t-1} & , \; i \notin S_t
\end{array}
\right.
\tag{11}
$$

where $\bar{\alpha}^{t-1} = \frac{1}{\lambda n} \sum_{i=1}^{n} A_i \alpha_i^{t-1}$, $\theta$ is a constant defined in (7) and $S_t \sim \hat{S}$ is a random subset of $[n]$. In other words, at iteration $t$ we first set the primal variable $w^t$ to be a convex combination of its current value $w^{t-1}$ and a value reducing $GAP_g$ to zero: see (10). This is followed by adjusting a subset of dual variables corresponding to a randomly chosen set of examples $S_t$ such that for each example $i \in S_t$, the $i$-th dual variable $\alpha_i^t$ is set to be a convex combination of its current value $\alpha_i^{t-1}$ and a value reducing $GAP_{\phi_i}$ to zero, see (11).

## 2.2 Flexibility of Quartz

Clearly, there are many ways in which the distribution of $\hat{S}$ can be chosen, leading to numerous variants of Quartz. The convex combination constant $\theta$ used throughout the algorithm should be tuned according to (7) where $v_1, \ldots, v_n$ are constants satisfying (5). Note that the best possible $v$ is obtained by computing the maximal eigenvalue of the matrix $(A^\top A) \circ P$ where $\circ$ denotes the Hadamard (component-wise) product of matrices and $P \in \mathbb{R}^{N \times N}$ is an $n$-by-$n$ block matrix with all elements in block $(i, j)$ equal to $\mathbb{P}(i \in \hat{S}, j \in \hat{S})$, see [14]. However, the worst-case complexity of computing directly the maximal eigenvalue of $(A^\top A) \circ P$ amounts to $O(N^2)$, which requires unreasonable preprocessing time in the context of machine learning where $N$ is assumed to be very large. We now describe some examples of sampling $\hat{S}$ and show how to compute in less than two passes over the data the corresponding constants $v_1, \ldots, v_n$. More examples including distributed sampling are presented in the supplementary material.

**Serial sampling.** The most studied sampling in the literature on stochastic optimization is the *serial sampling*, which corresponds to the selection of a single block $i \in [n]$. That is, $|\hat{S}| = 1$ with probability 1. The name "serial" is pointing to the fact that a method using such a sampling will typically be a serial (as opposed to being parallel) method; updating a single block (dual variable) at a time. A serial sampling is uniquely characterized by the vector of probabilities $p = (p_1, \dots, p_n)$, where $p_i$ is defined by (6). For serial sampling $\hat{S}$, it is easy to see that (5) is satisfied for

$$v_i = L_i \stackrel{\text{def}}{=} \lambda_{\max}(A_i^\top A_i), \quad i \in [n], \tag{12}$$

where $\lambda_{\max}(\cdot)$ denotes the maximal eigenvalue.

**Standard mini-batching.** We now consider $\hat{S}$ which selects subsets of $[n]$ of cardinality $\tau$, uniformly at random. In the terminology established in [17], such $\hat{S}$ is called $\tau$-nice. This sampling satisfies $p_i = p_j$ for all $i, j \in [n]$; and hence it is uniform. This sampling is well suited for parallel computing. Indeed, Quartz could be implemented as follows. If we have $\tau$ processors available, then at the beginning of iteration $t$ we can assign each block (dual variable) in $S_t$ to a dedicated processor. The processor assigned to $i$ would then compute $\Delta\alpha_i^t$ and apply the update. If all processors have fast access to the memory where all the data is stored, as is the case in a shared-memory multicore workstation, then this way of assigning workload to the individual processors does not cause any major problems. For $\tau$-nice sampling, (5) is satisfied for

$$v_i = \lambda_{\max}(M_i), \qquad M_i = \sum_{j=1}^d \left(1 + \frac{(\omega_j - 1)(\tau - 1)}{n - 1}\right) A_{ji}^\top A_{ji}, \qquad i \in [n], \tag{13}$$

where for each $j \in [d]$, $\omega_j$ is the number of nonzero blocks in the $j$-th row of matrix $A$, i.e.,

$$\omega_j \stackrel{\text{def}}{=} |\{i \in [n] : A_{ji} \neq 0\}|, \qquad j \in [d]. \tag{14}$$

Note that (13) follows from an extension of a formula given in [2] from $m = 1$ to $m \geq 1$.

## 3 Main Result

The complexity of our method is given by the following theorem. The proof can be found in the supplementary material.

**Theorem 2 (Main Result)** *Assume that $g$ is 1-strongly convex and that for each $i \in [n]$, $\phi_i$ is convex and $(1/\gamma)$-smooth. Let $\hat{S}$ be a proper sampling (Assumption 1) and $v_1, \dots, v_n$ be positive scalars satisfying (5). Then the sequence of primal and dual variables $\{w^t, \alpha^t\}_{t \geq 0}$ of Quartz (Algorithm 1) satisfies:*

$$\mathbb{E}[P(w^t) - D(\alpha^t)] \leq (1 - \theta)^t (P(w^0) - D(\alpha^0)), \tag{15}$$

*where $\theta$ is defined in (7). In particular, if we fix $\epsilon \leq P(w^0) - D(\alpha^0)$, then for*

$$T \geq \max_i \left(\frac{1}{p_i} + \frac{v_i}{p_i \lambda \gamma n}\right) \log\left(\frac{P(w^0) - D(\alpha^0)}{\epsilon}\right) \quad \Rightarrow \quad \mathbb{E}[P(w^T) - D(\alpha^T)] \leq \epsilon. \tag{16}$$

In order to put the above result into context, in the rest of this section we will specialize the above result to two special samplings: a serial sampling, and the $\tau$-nice sampling.

### 3.1 Quartz with serial sampling

When $\hat{S}$ is a serial sampling, we just need to plug (12) into (16) and derive the bound

$$T \geq \max_i \left(\frac{1}{p_i} + \frac{L_i}{p_i \lambda \gamma n}\right) \log\left(\frac{P(w^0) - D(\alpha^0)}{\epsilon}\right) \quad \Longrightarrow \quad \mathbb{E}[P(w^T) - D(\alpha^T)] \leq \epsilon. \tag{17}$$

If in addition, $\hat{S}$ is uniform, then $p_i = 1/n$ for all $i \in [n]$ and we refer to this special case of Quartz as Quartz-U. By replacing $p_i = 1/n$ in (17) we obtain directly the complexity of Quartz-U:

$$T \geq \left(n + \frac{\max_i L_i}{\lambda \gamma}\right) \log\left(\frac{P(w^0) - D(\alpha^0)}{\epsilon}\right) \quad \Longrightarrow \quad \mathbb{E}[P(w^T) - D(\alpha^T)] \leq \epsilon. \tag{18}$$

Otherwise, we can seek to maximize the right-hand side of the inequality in (17) with respect to the sampling probability $p$ to obtain the best bound. A simple calculation reveals that the optimal probability is given by:

$$\mathbb{P}(\hat{S} = \{i\}) = p_i^* \stackrel{\text{def}}{=} (L_i + \lambda\gamma n)/\sum_{i=1}^n (L_i + \lambda\gamma n). \tag{19}$$

We shall call Quartz-IP the algorithm obtained by using the above serial sampling probability. The following complexity result of Quartz-IP can be derived easily by plugging (19) into (17):

$$T \geq \left(n + \frac{\sum_{i=1}^n L_i}{n\lambda\gamma}\right) \log\left(\frac{P(w^0) - D(\alpha^0)}{\epsilon}\right) \quad \Longrightarrow \quad \mathbb{E}[P(w^T) - D(\alpha^T)] \leq \epsilon. \tag{20}$$

Note that in contrast with the complexity result of Quartz-U (18), we now have dependence on the *average* of the eigenvalues $L_i$.

**Quartz-U vs Prox-SDCA.** Quartz-U should be compared to Proximal Stochastic Dual Coordinate Ascent (Prox-SDCA) [22, 20]. Indeed, the dual update of Prox-SDCA takes exactly the same form of Quartz-U[1], see (11). The main difference is how the primal variable $w^t$ is updated: while Quartz performs the update (10), Prox-SDCA (see also [24, 5]) performs the more aggressive update $w^t = \nabla g^*(\bar{\alpha}^{t-1})$ and the complexity result of Prox-SDCA is as follows:

$$T \geq \left(n + \frac{\max_i L_i}{\lambda\gamma}\right) \log\left(\left(n + \frac{\max_i L_i}{\lambda\gamma}\right)\left(\frac{D(\alpha^*) - D(\alpha^0)}{\epsilon}\right)\right) \Rightarrow \mathbb{E}[P(w^T) - D(\alpha^T)] \leq \epsilon, \tag{21}$$

where $\alpha^*$ is the dual optimal solution. Notice that the dominant terms in (18) and (21) exactly match, although our logarithmic term is better and simpler. This is due to a direct bound on the decrease of the primal-dual error of Quartz, without the need to first analyze the dual error, in contrast to the typical approach for most of the dual coordinate ascent methods [22, 23, 20, 21, 9].

**Quartz-IP vs Iprox-SDCA.** The importance sampling (19) was previously used in the algorithm Iprox-SDCA [27], which extends Prox-SDCA to non-uniform serial sampling. The complexity of Quartz-IP (20) should then be compared with the following complexity result of Iprox-SDCA [27]:

$$T \geq \left(n + \frac{\sum_{i=1}^n L_i}{n\lambda\gamma}\right) \log\left(\left(n + \frac{\sum_{i=1}^n L_i}{n\lambda\gamma}\right)\left(\frac{D(\alpha^*) - D(\alpha^0)}{\epsilon}\right)\right) \Rightarrow \mathbb{E}[P(w^T) - D(\alpha^T)] \leq \epsilon. \tag{22}$$

Again, the dominant terms in (20) and (22) exactly match but our logarithmic term is smaller.

### 3.2 Quartz with $\tau$-nice Sampling (standard mini-batching)

We now specialize Theorem 2 to the case of the $\tau$-nice sampling. We define $\tilde{w}$ such that:

$$\max_i \lambda_{\max}\left(\sum_{j=1}^d \left(1 + \frac{(\omega_j - 1)(\tau - 1)}{n-1}\right) A_{ji}^\top A_{ji}\right) = \left(1 + \frac{(\tilde{\omega} - 1)(\tau - 1)}{n-1}\right) \max_i L_i$$

It is clear that $1 \leq \tilde{w} \leq \max_j w_j \leq n$ and can be considered as a measure of the *density* of the data. By plugging (13) into (16) we obtain directly the following corollary.

**Corollary 3** *Assume $\hat{S}$ is the $\tau$-nice sampling and $v$ is chosen as in* (13). *If we let $\epsilon \leq P(w^0) - D(\alpha^0)$ and*

$$T \geq \left(\frac{n}{\tau} + \frac{\left(1 + \frac{(\tilde{\omega} - 1)(\tau - 1)}{n-1}\right)\max_i L_i}{\lambda\gamma\tau}\right) \log\left(\frac{P(w^0) - D(\alpha^0)}{\epsilon}\right) \Rightarrow \mathbb{E}[P(w^T) - D(\alpha^T)] \leq \epsilon. \tag{23}$$

Let us now have a detailed look at the above result, especially in terms of how it compares with the serial uniform case (18). For fully sparse data, we get *perfect linear speedup*: the bound in (23) is a $1/\tau$ fraction of the bound in (18). For fully dense data, the condition number ($\kappa \stackrel{\text{def}}{=} \max_i L_i/(\lambda\gamma)$) is unaffected by mini-batching. For general data, the behaviour of Quartz with $\tau$-nice sampling interpolates these two extreme cases. It is important to note that regardless of the condition number $\kappa$, as long as $\tau \leq 1 + (n-1)/(\tilde{w}-1)$ the bound in (23) is at most a $2/\tau$ fraction of the bound in (18). Hence, for sparser problems, Quartz can achieve linear speedup for larger mini-batch sizes.

### 3.3 Quartz vs existing primal-dual mini-batch methods

We now compare the above result with existing mini-batch stochastic dual coordinate ascent methods. The mini-batch variants of SDCA, to which Quartz with $\tau$-nice sampling can be naturally compared, have been proposed and analyzed previously in [23], [21] and [26]. In [23], the authors proposed to use a so-called *safe mini-batching*, which is precisely equivalent to finding the stepsize parameter $v$ satisfying (5) (in the special case of $\tau$-nice sampling). However, they only analyzed the case where the functions $\{\phi_i\}_i$ are non-smooth. In [21], the authors studied accelerated mini-batch SDCA (ASDCA), specialized to the case when the regularizer $g$ is the squared L2 norm. They showed that the complexity of ASDCA interpolates between that of SDCA and accelerated gradient descent (AGD) [13] through varying the mini-batch size $\tau$. In [26], the authors proposed a mini-batch extension of their stochastic primal-dual coordinate algorithm (SPDC). Both ASDCA and SPDC reach the same complexity as AGD when the mini-batch size equals to $n$, thus should be considered as accelerated algorithms [2]. The complexity bounds for all these algorithms are summarized in Table 1. In Table 2 we compare the complexities of SDCA, ASDCA, SPDC and Quartz in several regimes.

| Algorithm | Iteration complexity | $g$ |
|---|---|---|
| SDCA [22] | $n + \frac{1}{\lambda\gamma}$ | $\frac{1}{2}\|\cdot\|^2$ |
| ASDCA [21] | $4 \times \max\left\{\frac{n}{\tau}, \sqrt{\frac{n}{\lambda\gamma\tau}}, \frac{1}{\lambda\gamma\tau}, \frac{n^{\frac{1}{3}}}{(\lambda\gamma\tau)^{\frac{2}{3}}}\right\}$ | $\frac{1}{2}\|\cdot\|^2$ |
| SPDC [26] | $\frac{n}{\tau} + \sqrt{\frac{n}{\lambda\gamma\tau}}$ | general |
| **Quartz with $\tau$-nice sampling** | $\frac{n}{\tau} + \left(1 + \frac{(\tilde{\omega}-1)(\tau-1)}{n-1}\right)\frac{1}{\lambda\gamma\tau}$ | general |

Table 1: Comparison of the iteration complexity of several primal-dual algorithms performing stochastic coordinate ascent steps in the dual using a mini-batch of examples of size $\tau$ (with the exception of SDCA, which is a serial method using $\tau = 1$.

| Algorithm | $\gamma\lambda n = \Theta(\frac{1}{\sqrt{n}})$ | $\gamma\lambda n = \Theta(1)$ | $\gamma\lambda n = \Theta(\tau)$ | $\gamma\lambda n = \Theta(\sqrt{n})$ |
|---|---|---|---|---|
| SDCA [22] | $n^{3/2}$ | $n$ | $n$ | $n$ |
| ASDCA [21] | $n^{3/2}/\tau + n^{5/4}/\sqrt{\tau} + n^{4/3}/\tau^{2/3}$ | $n/\sqrt{\tau}$ | $n/\tau$ | $n/\tau + n^{3/4}/\sqrt{\tau}$ |
| SPDC [26] | $n^{5/4}/\sqrt{\tau}$ | $n/\sqrt{\tau}$ | $n/\tau$ | $n/\tau + n^{3/4}/\sqrt{\tau}$ |
| **Quartz ($\tau$-nice)** | $n^{3/2}/\tau + \tilde{\omega}\sqrt{n}$ | $n/\tau + \tilde{\omega}$ | $n/\tau$ | $n/\tau + \tilde{\omega}/\sqrt{n}$ |

Table 2: Comparison of leading factors in the complexity bounds of several methods in 5 regimes.

Looking at Table 2, we see that in the $\gamma\lambda n = \Theta(\tau)$ regime (i.e., if the condition number is $\kappa = \Theta(n/\tau)$), Quartz matches the linear speedup (when compared to SDCA) of ASDCA and SPDC. When the condition number is roughly equal to the sample size ($\kappa = \Theta(n)$), then Quartz does better than both ASDCA and SPDC as long as $n/\tau + \tilde{\omega} \leq n/\sqrt{\tau}$. In particular, this is the case when the data is sparse: $\tilde{\omega} \leq n/\sqrt{\tau}$. If the data is even more sparse (and in many big data applications one has $\tilde{\omega} = O(1)$) and we have $\tilde{\omega} \leq n/\tau$, then Quartz significantly outperforms both ASDCA and SPDC. Note that Quartz can be better than both ASDCA and SPDC even in the domain of accelerated methods, that is, when the condition number is larger than the number of examples: $\kappa = 1/(\gamma\lambda) \geq n$. Indeed, we have the following result:

**Proposition 4** *Assume that $n\lambda\gamma \leq 1$ and that $\max_i L_i = 1$. If the data is sufficiently sparse so that*

$$\lambda\gamma\tau n \geq \left(1 + n\lambda\gamma + \frac{(\tilde{\omega}-1)(\tau-1)}{n-1}\right)^2, \tag{24}$$

*then the iteration complexity (in $\tilde{O}$ order) of Quartz is better than that of ASDCA and SPDC.*

The result can be interpreted as follows: if $n \leq \kappa \leq \tau n/(1 + n/\kappa)^2$ (that is, $\tau \geq \lambda\gamma\tau n \geq (1 + n\lambda\gamma)^2$), then there are sparse-enough problems for which Quartz is better than both ASDCA and SPDC.

# 4 Experimental Results

In this section we demonstrate how Quartz specialized to different samplings compares with other methods. All of our experiments are performed with $m = 1$, for smoothed hinge-loss functions $\{\phi_i\}$ with $\gamma = 1$ and squared L2-regularizer $g$, see [20]. The experiments were performed on the three datasets reported in Table 3, and three randomly generated large dataset [12] with $n = 100,000$ examples, $d = 100,000$ features with different sparsity. In Figure 1 we compare Quartz specialized to serial sampling and for both uniform and optimal sampling with Prox-SDCA and Iprox-SDCA, previously discussed in Section 3.1, on three datasets. Due to the conservative primal date in Quartz, Quartz-U appears to be slower than Prox-SDCA in practice. Nevertheless, in all the experiments, Quartz-IP shows almost identical convergence behaviour to that of Iprox-SDCA. In Figure 2 we compare Quartz specialized to $\tau$-nice sampling with mini-batch SPDC for different values of $\tau$, in the domain of accelerated methods ($\kappa = 10n$). The datasets are randomly generated following [13, Section 6]. When $\tau = 1$, it is clear that SPDC outperforms Quartz as the condition number is larger than $n$. However, as $\tau$ increases, the number of data processed by SPDC is increased by $\sqrt{\tau}$ as predicted by its theory but the number of data processed by Quartz remains almost the same by taking advantage of the large sparsity of the data. Hence, *Quartz is much better in the large $\tau$ regime.*

| Dataset | # Training size $n$ | # features $d$ | Sparsity (# nnz/$(nd)$) |
|---------|---------------------|----------------|--------------------------|
| cov1    | 522,911             | 54             | 22.22%                   |
| w8a     | 49,749              | 300            | 3.91%                    |
| ijcnn1  | 49,990              | 22             | 59.09%                   |

Table 3: Datasets used in our experiments.

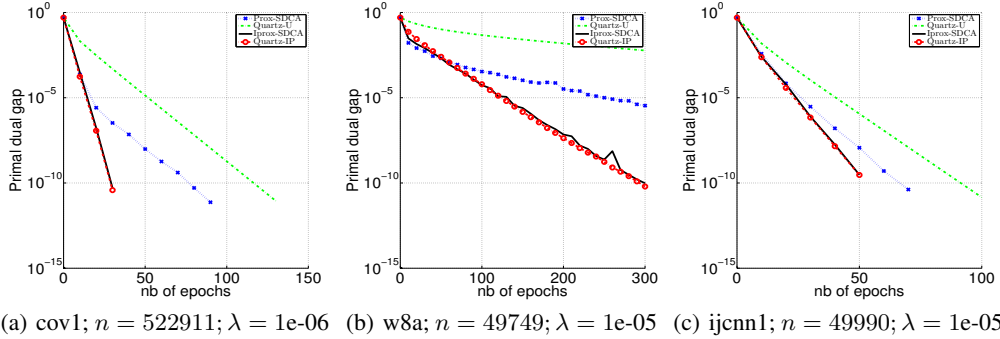

(a) cov1; $n = 522911$; $\lambda = $ 1e-06    (b) w8a; $n = 49749$; $\lambda = $ 1e-05    (c) ijcnn1; $n = 49990$; $\lambda = $ 1e-05

Figure 1: Comparison of Quartz-U (uniform sampling), Quartz-IP (optimal importance sampling), Prox-SDCA (uniform sampling) and Iprox-SDCA (optimal importance sampling).

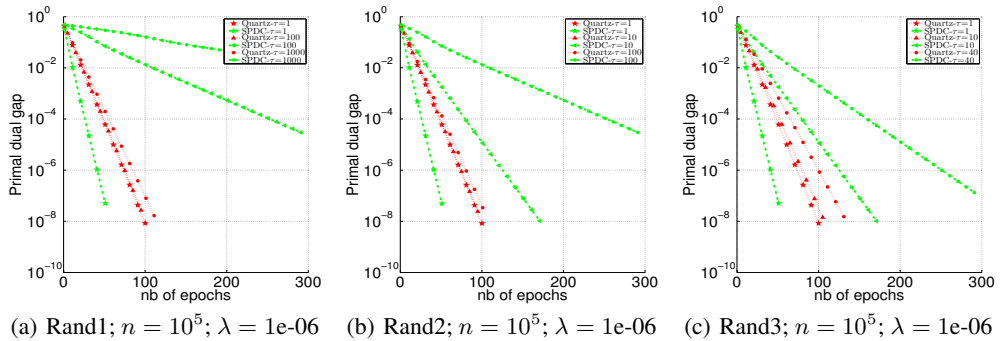

(a) Rand1; $n = 10^5$; $\lambda = $ 1e-06    (b) Rand2; $n = 10^5$; $\lambda = $ 1e-06    (c) Rand3; $n = 10^5$; $\lambda = $ 1e-06

Figure 2: Comparison of Quartz with SPDC for different mini-batch size $\tau$ in the regime $\kappa = 10n$. The three random datasets Random1, Random2 and Random2 have respective sparsity 0.01%, 0.1% and 1%.

## Footnotes

[1]In [20] the authors proposed five options of dual updating rule. Our dual updating formula (11) should be compared with option V in Prox-SDCA. For the same reason as given in the beginning of [20, Appendix A.], Quartz implemented with the same other four options achieves the same complexity result as Theorem 2.

[2]APCG [9] also reaches accelerated convergence rate but was not proposed in the mini-batch setting.

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
