[Supplementary Material]

# Quartz
# Supplementary material

In this document the reader can find the proof of Theorem 2 in Section 5, the proof of Proposition 4 in Secion 6, two more examples of sampling in Section 7 and the complexity results of Quartz specialzed in these two samplings in Section 8.

## 5 Proof of Theorem 2

In this section we prove our main result (Theorem 2). In order to make the analysis more transparent, we will first establish three auxiliary results.

### 5.1 Three lemmas

**Lemma 5.1** *Function $f : \mathbb{R}^N \to \mathbb{R}$ defined in (3) satisfies the following inequality:*

$$f(\alpha + h) \leq f(\alpha) + \langle \nabla f(\alpha), h \rangle + \frac{1}{2\lambda n^2} h^\top A^\top A h, \qquad \forall \alpha, h \in \mathbb{R}^N. \tag{25}$$

**Proof** Since $g$ is 1-strongly convex, $g^*$ is 1-smooth. Pick $\alpha, h \in \mathbb{R}^N$. Since, $f(\alpha) = \lambda g^*(\frac{1}{\lambda n} A\alpha)$, we have

$$
\begin{aligned}
f(\alpha + h) = \lambda g^* \left( \tfrac{1}{\lambda n} A\alpha + \tfrac{1}{\lambda n} Ah \right) &\leq \lambda \left( g^* \left( \tfrac{1}{\lambda n} A\alpha \right) + \left\langle \nabla g^* \left( \tfrac{1}{\lambda n} A\alpha \right), \tfrac{1}{\lambda n} Ah \right\rangle + \tfrac{1}{2} \left\| \tfrac{1}{\lambda n} Ah \right\|^2 \right) \\
&= f(\alpha) + \langle \nabla f(\alpha), h \rangle + \tfrac{1}{2\lambda n^2} h^\top A^\top A h.
\end{aligned}
$$

∎

For $s = (s_1, \ldots, s_n) \in \mathbb{R}^N$, $h = (h_1, \ldots, h_n) \in \mathbb{R}^N$, where $s_i, h_i \in \mathbb{R}^m$ for all $i$, we will for convenience write

$$\langle s, h \rangle_p = \sum_{i=1}^n p_i \langle s_i, h_i \rangle,$$

where $p = (p_1, \ldots, p_n)$ and $p_i = \mathbb{P}(i \in \hat{S})$ for $i \in [n]$.

In the next lemma we give an expected separable overapproximation of the convex function $-D$.

**Lemma 5.2** *If $\hat{S}$ and $v \in \mathbb{R}^n$ satisfy (5), then for all $\alpha, h \in \mathbb{R}^N$, the following holds:*

$$
\begin{aligned}
&\mathbb{E}[-D(\alpha + h_{[\hat{S}]})] \\
&\leq f(\alpha) + \langle \nabla f(\alpha), h \rangle_p + \frac{1}{2\lambda n^2} \|h\|_{p \cdot v}^2 + \frac{1}{n} \sum_{i=1}^n \left[ (1 - p_i) \phi_i^*(-\alpha_i) + p_i \phi_i^*(-\alpha_i - h_i) \right].
\end{aligned} \tag{26}
$$

**Proof** By definition of $D$, we have

$$-D(\alpha + h_{[\hat{S}]}) \overset{(2)}{=} f(\alpha + h_{[\hat{S}]}) + \psi(\alpha + h_{[\hat{S}]}),$$

where $f$ and $\psi$ are defined in (3). Now we apply Lemma 5.1 and (5) to bound the first term:

$$
\begin{aligned}
\mathbb{E}[f(\alpha + h_{[\hat{S}]})] &\overset{(25)}{\leq} \mathbb{E}[f(\alpha) + \langle \nabla f(\alpha), h_{[\hat{S}]}\rangle + \frac{1}{2\lambda n^2} h_{[\hat{S}]}^\top A^\top A h_{[\hat{S}]}] \\
&\overset{(5)}{\leq} f(\alpha) + \mathbb{E}[\langle \nabla f(\alpha), h_{[\hat{S}]}\rangle] + \frac{1}{2\lambda n^2}\|h\|_{p \cdot v}^2 \\
&= f(\alpha) + \langle \nabla f(\alpha), h\rangle_p + \frac{1}{2\lambda n^2}\|h\|_{p \cdot v}^2.
\end{aligned}
$$

Moreover, since $\psi$ is block separable, we can write

$$
\begin{aligned}
\mathbb{E}[\psi(\alpha + h_{[\hat{S}]})] &\overset{(3)}{=} \frac{1}{n}\sum_{i=1}^n \left[ \mathbb{P}(i \notin \hat{S})\phi_i^*(-\alpha_i) + \mathbb{P}(i \in \hat{S})\phi_i^*(-\alpha_i - h_i) \right] \\
&= \frac{1}{n}\sum_{i=1}^n \left[(1 - p_i)\phi_i^*(-\alpha_i) + p_i\phi_i^*(-\alpha_i - h_i)\right].
\end{aligned}
$$

∎

Our last auxiliary result is a technical lemma for further bounding the right hand side in Lemma 5.2.

**Lemma 5.3** *Suppose that $\hat{S}$ and $v \in \mathbb{R}^n$ satisfy (5). Fixing $\alpha \in \mathbb{R}^N$ and $w \in \mathbb{R}^d$, let $h \in \mathbb{R}^N$ be defined by:*
$$
h_i = -\theta p_i^{-1}(\alpha_i + \nabla\phi_i(A_i^\top w)), \quad i \in [n],
$$
*where $\theta$ be as in (7). Then*

$$
\begin{aligned}
&f(\alpha) + \langle \nabla f(\alpha), h\rangle_p + \frac{1}{2\lambda n^2}\|h\|_{p\cdot v}^2 + \frac{1}{n}\sum_{i=1}^n [(1 - p_i)\phi_i^*(-\alpha_i) + p_i\phi_i^*(-\alpha_i - h_i)] \\
&\leq -(1 - \theta)D(\alpha) - \theta\lambda g(\nabla g^*(\bar{\alpha})) - \frac{1}{n}\sum_{i=1}^n \langle \theta\nabla g^*(\bar{\alpha}), A_i\nabla\phi_i(A_i^\top w)\rangle + \frac{\theta}{n}\sum_{i=1}^n \phi_i^*(\nabla\phi_i(A_i^\top w)),
\end{aligned}
\tag{27}
$$

*where $\bar{\alpha} = \frac{1}{\lambda n}A\alpha$.*

**Proof** Recall from (3) that $f(\alpha) = \lambda g^*(\bar{\alpha})$ and hence $\nabla f(\alpha) = \frac{1}{n}A^\top \nabla g^*(\bar{\alpha})$. Thus,

$$
\begin{aligned}
&f(\alpha) + \langle \nabla f(\alpha), h\rangle_p + \frac{1}{2\lambda n^2}\|h\|_{p\cdot v}^2 \\
&= \lambda g^*(\bar{\alpha}) - \sum_{i=1}^n p_i \langle \frac{1}{n}A_i^\top \nabla g^*(\bar{\alpha}), \theta p_i^{-1}(\alpha_i + \nabla\phi_i(A_i^\top w))\rangle + \frac{1}{2\lambda n^2}\|h\|_{p\cdot v}^2 \\
&= (1 - \theta)\lambda g^*(\bar{\alpha}) + \theta\lambda(g^*(\bar{\alpha}) - \langle \nabla g^*(\bar{\alpha}), \bar{\alpha}\rangle) \\
&\quad - \frac{1}{n}\sum_{i=1}^n \langle \theta\nabla g^*(\bar{\alpha}), A_i\nabla\phi_i(A_i^\top w)\rangle + \frac{1}{2\lambda n^2}\|h\|_{p\cdot v}^2.
\end{aligned}
\tag{28}
$$

Since the functions $\phi_i$ are $(1/\gamma)$-smooth, the conjugate functions $\phi_i^*$ must be $\gamma$-strongly convex. Therefore,

$$
\begin{aligned}
&\phi_i^*(-\alpha_i - h_i) \\
&= \phi_i^*(-(1 - \theta p_i^{-1})\alpha_i + \theta p_i^{-1}\nabla\phi_i(A_i^\top w)) \\
&\leq (1 - \theta p_i^{-1})\phi_i^*(-\alpha_i) + \theta p_i^{-1}\phi_i^*(\nabla\phi_i(A_i^\top w)) - \frac{\gamma\theta p_i^{-1}(1 - \theta p_i^{-1})}{2}\|\alpha_i + \nabla\phi_i(A_i^\top w)\|^2 \\
&= (1 - \theta p_i^{-1})\phi_i^*(-\alpha_i) + \theta p_i^{-1}\phi_i^*(\nabla\phi_i(A_i^\top w)) - \frac{\gamma p_i(1 - \theta p_i^{-1})}{2\theta}\|h_i\|^2,
\end{aligned}
\tag{29}
$$

and we can write

$$\frac{1}{n}\sum_{i=1}^{n}[(1-p_i)\phi_i^*(-\alpha_i) + p_i\phi_i^*(-\alpha_i - h_i)]$$

$$\overset{(29)}{\leq} (1-\theta)\psi(\alpha) + \frac{\theta}{n}\sum_{i=1}^{n}(\phi_i^*(\nabla\phi_i(A_i^\top w))) - \frac{1}{2\lambda n^2}\sum_{i=1}^{n}\frac{n\lambda\gamma p_i^2(1-\theta p_i^{-1})}{\theta}\|h_i\|^2. \qquad (30)$$

Then by combining (28) and (30) we get:

$$f(\alpha) + \langle\nabla f(\alpha), h\rangle_p + \frac{1}{2\lambda n^2}\|h\|_{p\cdot v}^2 + \frac{1}{n}\sum_{i=1}^{n}[(1-p_i)\phi_i^*(-\alpha_i) + p_i\phi_i^*(-\alpha_i - h_i)]$$

$$\leq -(1-\theta)D(\alpha) - \theta\lambda g(\nabla g^*(\bar{\alpha})) - \frac{1}{n}\sum_{i=1}^{n}\langle\theta\nabla g^*(\bar{\alpha}), A_i\nabla\phi_i(A_i^\top w)\rangle + \frac{\theta}{n}\sum_{i=1}^{n}\phi_i^*(\nabla\phi_i(A_i^\top w))$$

$$+ \frac{1}{2\lambda n^2}\sum_{i=1}^{n}\left(p_i v_i - \frac{n\lambda\gamma p_i^2(1-\theta p_i^{-1})}{\theta}\right)\|h_i\|^2.$$

It remains to notice that for $\theta$ defined in (7), we have:

$$p_i v_i \leq \frac{n\lambda\gamma p_i^2(1-\theta p_i^{-1})}{\theta}, \quad \forall i \in [n].$$

∎

## 5.2 Proof of Theorem 2

Let $t \geq 1$. Define $h^t = (h_1^t, \ldots, h_n^t) \in \mathbb{R}^N$ by:

$$h_i^t = -\theta p_i^{-1}(\alpha_i^{t-1} + \nabla\phi_i(A_i^\top w^t)), \quad i \in [n]$$

and $\kappa^t = (\kappa_1^t, \cdots, \kappa_n^t)$ by:

$$\kappa_i^t = \arg\max_{\Delta\in\mathbb{R}^m}\left[-\phi_i^*(-(\alpha_i^{t-1} + \Delta)) - \nabla g^*(\bar{\alpha}^{t-1})^\top A_i\Delta - \frac{v_i\|\Delta\|^2}{2\lambda n}\right], \quad \forall i \in [n].$$

If we use Option I in Algorithm 1, then $\alpha^t = \alpha^{t-1} + \kappa_{[\hat{S}]}^t$. If we use Option II in Algorithm 1, then we have $\alpha^t = \alpha^{t-1} + h_{[\hat{S}]}^t$. In both cases, by Lemma 5.2:

$$\mathbb{E}_t[-D(\alpha^t)]$$

$$\leq f(\alpha^{t-1}) + \langle\nabla f(\alpha^{t-1}), h^t\rangle_p + \frac{1}{2\lambda n^2}\|h^t\|_{p\cdot v}^2 + \frac{1}{n}\sum_{i=1}^{n}\left[(1-p_i)\phi_i^*(-\alpha_i^{t-1}) + p_i\phi_i^*(-\alpha_i^{t-1} - h_i^t)\right].$$

We now apply Lemma 5.3 to further bound the last term and obtain:

$$\mathbb{E}_t[-D(\alpha^t)] \leq -(1-\theta)D(\alpha^{t-1}) - \theta\lambda g(\nabla g^*(\bar{\alpha}^{t-1}))$$

$$- \frac{1}{n}\sum_{i=1}^{n}\langle\theta\nabla g^*(\bar{\alpha}^{t-1}), A_i\nabla\phi_i(A_i^\top w^t)\rangle + \frac{\theta}{n}\sum_{i=1}^{n}\phi_i^*(\nabla\phi_i(A_i^\top w^t)). \qquad (31)$$

By convexity of $g$,

$$P(w^t) = \frac{1}{n}\sum_{i=1}^{n}\phi_i(A_i^\top w^t) + \lambda g((1-\theta)w^{t-1} + \theta\nabla g^*(\bar{\alpha}^{t-1}))$$

$$\leq \frac{1}{n}\sum_{i=1}^{n}\phi_i(A_i^\top w^t) + (1-\theta)\lambda g(w^{t-1}) + \theta\lambda g(\nabla g^*(\bar{\alpha}^{t-1})). \qquad (32)$$

By combining (31) and (32) we get:

$$
\begin{aligned}
\mathbb{E}_t[P(w^t) - D(\alpha^t)] \leq & \frac{1}{n}\sum_{i=1}^n \phi_i(A_i^\top w^t) + (1-\theta)\lambda g(w^{t-1}) - (1-\theta)D(\alpha^{t-1}) \\
& - \frac{1}{n}\sum_{i=1}^n \langle \theta \nabla g^*(\bar{\alpha}^{t-1}), A_i \nabla \phi_i(A_i^\top w^t)\rangle + \frac{\theta}{n}\sum_{i=1}^n \phi_i^*(\nabla \phi_i(A_i^\top w^t)) \\
= & (1-\theta)(P(w^{t-1}) - D(\alpha^{t-1})) + \frac{1}{n}\sum_{i=1}^n (\phi_i(A_i^\top w^t) - (1-\theta)\phi_i(A_i^\top w^{t-1})) \\
& - \frac{1}{n}\sum_{i=1}^n \langle \theta \nabla g^*(\bar{\alpha}^{t-1}), A_i \nabla \phi_i(A_i^\top w^t)\rangle + \frac{\theta}{n}\sum_{i=1}^n \phi_i^*(\nabla \phi_i(A_i^\top w^t)).
\end{aligned} \tag{33}
$$

Note that $\theta \nabla g^*(\bar{\alpha}^{t-1}) = w^t - (1-\theta)w^{t-1}$ and $\phi_i^*(\nabla \phi_i(A_i^\top w^t)) = \langle \nabla \phi_i(A_i^\top w^t), A_i^\top w^t\rangle - \phi_i(A_i^\top w^t)$. Finally, we plug these two inequalities into (33) and obtain:

$$
\begin{aligned}
\mathbb{E}_t[P(w^t) - D(\alpha^t)] \leq & (1-\theta)(P(w^{t-1}) - D(\alpha^{t-1})) + \frac{1}{n}\sum_{i=1}^n (\phi_i(A_i^\top w^t) - (1-\theta)\phi_i(A_i^\top w^{t-1})) \\
& - \frac{1}{n}\sum_{i=1}^n \langle A_i^\top w^t - (1-\theta)A_i^\top w^{t-1}, \nabla \phi_i(A_i^\top w^t)\rangle \\
& + \frac{\theta}{n}\sum_{i=1}^n \left(\langle \nabla \phi_i(A_i^\top w^t), A_i^\top w^t\rangle - \phi_i(A_i^\top w^t)\right) \\
= & (1-\theta)(P(w^{t-1}) - D(\alpha^{t-1})) + \frac{1-\theta}{n}\sum_{i=1}^n (\phi_i(A_i^\top w^t) - \phi_i(A_i^\top w^{t-1})) \\
& - \frac{1-\theta}{n}\sum_{i=1}^n \langle A_i^\top w^t - A_i^\top w^{t-1}, \nabla \phi_i(A_i^\top w^t)\rangle \\
= & (1-\theta)(P(w^{t-1}) - D(\alpha^{t-1})) \\
& + \frac{1-\theta}{n}\sum_{i=1}^n \left[\phi_i(A_i^\top w^t) - \phi_i(A_i^\top w^{t-1}) + \langle A_i^\top w^{t-1} - A_i^\top w^t, \nabla \phi_i(A_i^\top w^t)\rangle\right] \\
\leq & (1-\theta)(P(w^{t-1}) - D(\alpha^{t-1})),
\end{aligned}
$$

where the last inequality follows from the convexity of $\phi_i$.

## 6 Proof of Proposition 4

**Proof** [Proof of Proposition 4] As long as $\lambda \gamma \tau n \geq 1$, which holds under our assumption, the iteration complexity of ASDCA is:

$$
\tilde{O}\left(\max\left\{\frac{n}{\tau}, \sqrt{\frac{n}{\lambda \gamma \tau}}, \frac{1}{\lambda \gamma \tau}, \frac{n^{\frac{1}{3}}}{(\lambda \gamma \tau)^{\frac{2}{3}}}\right\}\right) = \tilde{O}\left(\sqrt{\frac{n}{\lambda \gamma \tau}}\right).
$$

which is already less than that of SPDC. Moreover,

$$
\sqrt{\frac{n}{\lambda \gamma \tau}} \overset{(24)}{\geq} \frac{1 + n\lambda \gamma + \frac{(\tau-1)(\tilde{\omega}-1)}{n-1}}{\lambda \gamma \tau} = \frac{n}{\tau} + \frac{1 + \frac{(\tau-1)(\tilde{\omega}-1)}{n-1}}{\lambda \gamma \tau}.
$$

∎

# 7  Two more samplings

## 7.1  Product sampling

In this section we give an example of a sampling $\hat{S}$ which can be both non-uniform and non-serial (i.e., for which $\mathbb{P}(|\hat{S}| = 1) \neq 1$). We make the following *group separability assumption*: there is a partition $X_1, \ldots, X_\tau$ of $[n]$ according to which the examples $\{A_i\}$ can be partitioned into $\tau$ groups such that no feature is shared by any two examples belonging to different groups.

Consider the following example with $m = 1, n = 5$ and $d = 4$:

$$A = [A_1, A_2, A_3, A_4, A_5] = \begin{pmatrix} 0 & 0 & 6 & 4 & 9 \\ 0 & 3 & 0 & 0 & 0 \\ 0 & 0 & 3 & 0 & 1 \\ 1 & 8 & 0 & 0 & 0 \end{pmatrix}$$

If we choose $\tau = 2$ and $X_1 = \{1, 2\}, X_2 = \{3, 4, 5\}$, then no row of $A$ has a nonzero in both a column belonging to $X_1$ and a column belonging to $X_2$.

With each $i \in [n]$ we now associate $l_i \in [\tau]$ such that $i \in X_{l_i}$ and define:

$$S \overset{\text{def}}{=} X_1 \times \cdots \times X_\tau.$$

The *product sampling* $\hat{S}$ is obtained by choosing $S \in \mathcal{S}$, uniformly at random; that is, via:

$$\mathbb{P}(\hat{S} = S) = \frac{1}{|\mathcal{S}|} = \frac{1}{\prod_{l=1}^\tau |X_l|}, \quad S \in \mathcal{S}. \tag{34}$$

Then $\hat{S}$ is proper and

$$p_i \overset{\text{def}}{=} \mathbb{P}(i \in \hat{S}) = \frac{\prod_{l \neq l_i} |X_l|}{|\mathcal{S}|} \overset{(34)}{=} \frac{1}{|X_{l_i}|}, \quad i \in [n]. \tag{35}$$

Hence the sampling is nonuniform as long as not all of the sets $X_l$ have the same cardinality. We next show that the product sampling $\hat{S}$ defined as above allows the same stepsize parameter $v_i$ as the serial uniform sampling.

**Lemma 7.1** *Under the group separability assumption,* (5) *is satisfied for the product sampling $\hat{S}$ and*

$$v_i = \lambda_{\max}(A_i^\top A_i), \quad i \in [n].$$

**Proof** For each $j \in [d]$, denote by $A_{j:}$ the $j$-th row of the matrix $A$ and $\Omega_j$ the column index set of nonzero blocks in $A_{j:}$: $\Omega_j \overset{\text{def}}{=} \{i \in [n] : A_{ji} \neq 0\}$. For each $l \in [\tau]$, define:

$$J_l \overset{\text{def}}{=} \{j \in [d] : \Omega_j \subset X_l\}. \tag{36}$$

In words, $J_l$ is the set of features associated with the examples in $X_l$. By the group separability assumption, $J_1, \ldots, J_\tau$ forms a partition of $[d]$, namely,

$$\bigcup_{l=1}^\tau J_l = [d]; \quad J_k \cap J_l = \emptyset, \quad \forall k \neq l \in [\tau]. \tag{37}$$

Thus,

$$A^\top A = \sum_{j=1}^d A_{j:}^\top A_{j:} \overset{(37)}{=} \sum_{l=1}^\tau \sum_{j \in J_l} A_{j:}^\top A_{j:}. \tag{38}$$

Now fix $l \in [\tau]$ and $j \in J_l$. For any $h \in \mathbb{R}^N$ we have:

$$\mathbb{E}[h_{[\hat{S}]}^\top A_{j:}^\top A_{j:} h_{[\hat{S}]}] = \sum_{i,i' \in [n]} h_i^\top A_{ji}^\top A_{ji'} h_{i'} \mathbb{P}(i \in \hat{S}, i' \in \hat{S}) = \sum_{i,i' \in \Omega_j} h_i^\top A_{ji}^\top A_{ji'} h_{i'} \mathbb{P}(i \in \hat{S}, i' \in \hat{S}).$$

Since $X_1, \ldots, X_\tau$ forms a partition of $[n]$, then any two indexes belonging to the same subset $X_l$ will never be selected simultaneously in $\hat{S}$, i.e.,

$$\mathbb{P}(i \in \hat{S}, i' \in \hat{S}) = \begin{cases} p_i & \text{if } i = i' \\ 0 & \text{if } i \neq i' \end{cases}, \quad \forall i, i' \in X_l.$$

Therefore,

$$\mathbb{E}[h_{[\hat{S}]} A_{j:}^\top A_{j:} h_{[\hat{S}]}] = \sum_{i \in \Omega_j} h_i^\top A_{ji}^\top A_{ji} h_i p_i = \sum_{i=1}^n h_i^\top A_{ji}^\top A_{ji} h_i p_i. \tag{39}$$

It follows from (38) and (39) that:

$$\mathbb{E}[\|A h_{[\hat{S}]}\|^2] = \mathbb{E}[h_{[\hat{S}]} A^\top A h_{[\hat{S}]}] = \sum_{l=1}^\tau \sum_{j \in J_l} \mathbb{E}[h_{[\hat{S}]} A_{j:}^\top A_{j:} h_{[\hat{S}]}] = \sum_{l=1}^\tau \sum_{j \in J_l} \sum_{i=1}^n h_i^\top A_{ji}^\top A_{ji} h_i p_i. \tag{40}$$

Hence, $\mathbb{E}[\|A h_{[\hat{S}]}\|^2] \overset{(37)}{=} \sum_{j=1}^d \sum_{i=1}^n h_i^\top A_{ji}^\top A_{ji} h_i p_i \leq \sum_{i=1}^n \lambda_{\max}(A_i^\top A_i) h_i^\top h_i p_i = \|h\|_{p \cdot v}^2.$ ∎

## 7.2 Distributed sampling

We now describe a sampling which is particularly suitable for a *distributed implementation of Quartz*. This sampling was first proposed in [RT13] and later used in [FQRT14], where the distributed coordinate descent algorithm Hydra and its accelerated variant Hydra$^2$ were proposed and analyzed, respectively. Both methods were shown to be able to scale up to huge problem sizes (tests were performed on problem sizes of several TB; and up 50 billion dual variables in size).

Consider a distributed computing environment with $c$ nodes/computers. For simplicity, assume that $n$ is an integer multiple of $c$ and let the blocks $\{1, 2, \ldots, n\}$ be partitioned into $c$ sets of equal size: $\mathcal{P}_1, \mathcal{P}_2, \ldots, \mathcal{P}_c$. We assign partition $\mathcal{P}_l$ to node $l$. The data $A_1, \ldots, A_n$ and the dual variables (blocks) $\alpha_1, \ldots, \alpha_n$ are partitioned accordingly and stored on the respective nodes.

At each iteration, all nodes $l \in \{1, \ldots, c\}$ in parallel pick a subset $\hat{S}_l$ of $\tau$ dual variables from those they own, i.e., from $\mathcal{P}_l$, uniformly at random. That is, each node locally performs a $\tau$-nice sampling, independently from the other nodes. Node $l$ computes the updates to the dual variables $\alpha_i$ corresponding to $i \in S_l$, and locally stores them. Hence, in a single distributed iteration, Quartz updates the dual variables belonging to the set $\hat{S} \overset{\text{def}}{=} \cup_{l=1}^c \hat{S}_l$. This defines a sampling, which we will call $(c, \tau)$-*distributed sampling*.

Of course, there are other important considerations pertaining to the distributed implementation of Quartz, but we do not discuss them here as the focus of this section is on the sampling. However, it is possible to design a distributed communication protocol for the update of the primal variable.

The following result gives a formula for admissible parameters $\{v_i\}$.

**Lemma 7.2 (compare with [FQRT14])** *If $\hat{S}$ is a $(c, \tau)$-distributed sampling, then (5) is satisfied for*

$$v_i = \lambda_{\max} \left( \sum_{j=1}^d \left( 1 + \frac{(\tau - 1)(\omega_j - 1)}{\max\left\{\frac{n}{c} - 1, 1\right\}} + \left( \frac{\tau c}{n} - \frac{\tau - 1}{\max\{\frac{n}{c} - 1, 1\}} \right) \frac{\omega_j' - 1}{\omega_j'} \omega_j \right) A_{ji}^\top A_{ji} \right), \quad i \in [n], \tag{41}$$

*where $\omega_j$ is the number of nonzero blocks in the $j$-th row of the matrix $A$, as defined previously in (14), and $\omega_j'$ is the number of partitions "active" at row $j$ of $A$, more precisely,*

$$\omega_j' \overset{\text{def}}{=} |\{l \in [c] : \{i \in \mathcal{P}_l : A_{ji} \neq 0\} \neq \emptyset\}|, \qquad j \in [d]. \tag{42}$$

**Proof** When $m = 1$, the result is equivalent to Theorem 4.1 in [FQRT14]. The extension to blocks ($m > 1$) is straightforward. ∎

The formula (13) is a special case of Lemma 7.2 when only a single node ($c = 1$) is used, in which case $\omega'_j = 1$ for all $j \in [d]$. Lemma 7.2 also improves the constants $\{v_i\}$ derived in [RT13], where instead of $\omega_j$ and $\omega'_j$ in (41) one has $\max_j \omega_j$ and $\max_j \omega'_j$.

Lemma 7.2 is expressed in terms of certain sparsity parameters associated with the data ($\{\omega_j\}$) and the partitioning ($\{\omega'_j\}$). However, it is possible to derive alternative ESO results for the $(c, \tau)$-distributed sampling. For instance, one can instead express the parameters $\{v_j\}$ without any sparsity assumptions, using only spectral properties of the data only. We have not included these results here, but in the $m = 1$ case such results have been derived in [FQRT14]. It is possible to adopt them to the $m > 1$ case as we have done it with Lemma 7.2.

# 8 Quartz specialized to product sampling and distributed sampling

## 8.1 Quartz with product sampling

In this section we apply Theorem 2 to the case when $\hat{S}$ is the product sampling (see the description in Section 7.1). All the notation we use here was established there.

**Corollary 1** *Under the group separability assumption, let $\hat{S}$ be the product sampling and let $v_i = \lambda_{\max}(A_i^\top A_i)$ for all $i \in [n]$. If we fix $\epsilon \leq P(w^0) - D(\alpha^0)$ and*

$$T \geq \max_i \left( |X_{l_i}| + \frac{\lambda_{\max}(A_i^\top A_i)|X_{l_i}|}{\lambda \gamma n} \right) \log \left( \frac{P(w^0) - D(\alpha^0)}{\epsilon} \right), \tag{43}$$

*then $\mathbb{E}[P(w^T) - D(\alpha^T)] \leq \epsilon$.*

**Proof** The proof follows directly from Theorem 2, Lemma 7.1 and (35). ∎

Recall from Section 7.1 that the product sampling $\hat{S}$ has cardinality $\tau \geq 1$ and is non-uniform as long as all the sets $\{X_1, \ldots, X_\tau\}$ do not have the same cardinality. To the best of our knowledge, Corollary 1 is the first *explicit* complexity bound of stochastic algorithm using non-serial and non-uniform sampling for composite convex optimization problem (the paper [RT15] only deals with smooth functions and the method is not primal-dual), albeit under the group separability assumption.

Let us compare the complexity bound (43) with that of serial uniform sampling (18):

$$\frac{n + \frac{\max_i \lambda_{\max}(A_i^\top A_i)}{\lambda \gamma}}{\max_i \left( |X_{l_i}| + \frac{\lambda_{\max}(A_i^\top A_i)|X_{l_i}|}{\lambda \gamma n} \right)} \geq \min_i \frac{n}{|X_{l_i}|}.$$

Hence the iteration bound of Quartz specialized to product sampling is at most a $\max_i |X_{l_i}|/n$ fraction of that of Quartz specialized to serial uniform sampling. The factor $\max_i |X_{l_i}|/n$ varies from $1/\tau$ to $1$, depending on the degree to which the partition $X_1, \ldots, X_\tau$ is balanced. A perfect linear speedup ($\max_i |X_{l_i}|/n = 1/\tau$) only occurs when the partition $X_1, \ldots, X_\tau$ is perfectly balanced (i.e., the set $X_l$ have the same cardinality), in which case the product sampling is uniform (recall the definition of uniformity we use in this paper: $\mathbb{P}(i \in \hat{S}) = \mathbb{P}(i' \in \hat{S})$ for all $i, i' \in [n]$). Note that if the partition is not perfectly but sufficiently so, then the factor $\max_i |X_{l_i}|/n$ will be close to the perfect linear speedup factor $1/\tau$.

## 8.2 Quartz with Distributed Sampling

In this section we apply Theorem 2 to the case when $\hat{S}$ is the $(c, \tau)$-distributed sampling; see the description of this sampling in Section 7.2.

**Corollary 2** *Assume that $\hat{S}$ is a $(c, \tau)$-distributed sampling and $v$ is chosen as in (41). If we let $\epsilon \leq P(w^0) - D(\alpha^0)$ and*

$$T \geq T(c, \tau) \times \log \left( \frac{P(w^0) - D(\alpha^0)}{\epsilon} \right), \tag{44}$$

*where*

$$T(c,\tau) \overset{def}{=} \frac{n}{c\tau} + \max_i \frac{\lambda_{\max}\left(\sum_{j=1}^d \left(1 + \frac{(\tau-1)(\omega_j-1)}{\max\{n/c-1,1\}} + \left(\frac{\tau c}{n} - \frac{\tau-1}{\max\{n/c-1,1\}}\right)\frac{\omega_j'-1}{\omega_j'}\omega_j\right) A_{ji}^\top A_{ji}\right)}{\lambda\gamma c\tau}, \qquad (45)$$

*then* $\mathbb{E}[P(w^T) - D(\alpha^T)] \leq \epsilon$.

**Proof** If $\hat{S}$ is a $(c,\tau)$-distributed sampling, then

$$p_i = \frac{c\tau}{n}, \qquad i \in [n].$$

It now only remains to combine Theorem 2 and Lemma 7.2. ∎

The expression (45) involves $\omega_j'$, which depends on the partitioning $\{\mathcal{P}_1, \mathcal{P}_2, \ldots, \mathcal{P}_c\}$ of the dual variable and the data. The following lemma says that the effect of the partition is negligible, and in fact vanishes as $\tau$ increases. It was proved in [FQRT14, Lemma 5.2].

**Lemma 8.1 ([FQRT14])** *If $n/c \geq 2$ and $\tau \geq 2$, then for all $j \in [d]$, we have*

$$\left(\frac{\tau c}{n} - \frac{\tau-1}{n/c-1}\right)\frac{\omega_j'-1}{\omega_j'}\omega_j \leq \frac{1}{\tau-1}\left(1 + \frac{(\tau-1)(\omega_j-1)}{n/c-1}\right).$$

According to this result, when each node owns at least two dual examples ($n/c \geq 2$) and picks and updates at least two examples in each iteration ($\tau \geq 2$), then

$$\begin{aligned}
T(c,\tau) &\leq \frac{n}{c\tau} + \left(1 + \frac{1}{\tau-1}\right)\frac{\max_i \lambda_{\max}\left(\sum_{j=1}^d\left(1 + \frac{(\tau-1)(\omega_j-1)}{n/c-1}\right)A_{ji}^\top A_{ji}\right)}{\lambda\gamma c\tau} \\
&= \frac{n}{c\tau} + \left(1 + \frac{1}{\tau-1}\right)\left(1 + \frac{(\tau-1)(\hat{\omega}-1)}{n/c-1}\right)\frac{\max_i \lambda_{\max}(A_i^\top A_i)}{\lambda\gamma c\tau},
\end{aligned} \qquad (46)$$

where $\hat{\omega} \in [1, n]$ is an *average sparsity measure* similar to that one we introduced in the study of $\tau$-nice sampling. This bound is similar to that we obtained for the $\tau$-nice sampling; and can be interpreted in an analogous way. Note that as the first term ($n$) receives perfect mini-batch scaling (it is divided by $c\tau$), while the condition number $\max_i \lambda_{\max}(A_i^\top A_i)/(\lambda\gamma)$ is divided by $c\tau$ but also multiplied by $\left(1 + \frac{1}{\tau-1}\right)\left(1 + \frac{(\tau-1)(\hat{\omega}-1)}{n/c-1}\right)$. However, this term is bounded by $2\hat{\omega}$, and hence if $\hat{\omega}$ is small, the condition number also receives a nearly perfect mini-batch scaling.

## 8.3 Quartz vs DiSDCA

A distributed variant of SDCA, named DiSDCA, has been proposed in [Yan13] and analyzed in [YZL13]. The authors of [Yan13] proposed a basic DiSDCA variant (which was analyzed) and a practical DiSDCA variant (which was not analyzed). The complexity of basic DiSDCA was shown to be:

$$\left(\frac{n}{c\tau} + \frac{\max_i \lambda_{\max}(A_i^\top A_i)}{\lambda\gamma}\right)\log\left(\frac{n}{c\tau} + \left(\frac{\max_i \lambda_{\max}(A_i^\top A_i)}{\lambda\gamma}\right) \cdot \frac{D(\alpha^*) - D(\alpha^0)}{\epsilon}\right), \qquad (47)$$

where $\alpha^*$ is an optimal dual solution. Note that this rate is much worse than our rate. Ignoring the logarithmic terms, while the first expression $n/(c\tau)$ is the same in both results, if we replace all $\omega_j$ by the upper bound $n$ and all $\omega_j'$ by the upper bound $c$ in (45), then

$$\begin{aligned}
T(c,\tau) &\leq \frac{n}{c\tau} + \left(\max_i \lambda_{\max}(A_i^\top A_i)\right) \cdot \frac{1 + \frac{(\tau-1)(n-1)}{\max(n/c-1)} + \left(\frac{\tau c}{n} - \frac{\tau-1}{\max(n/c-1,1)}\right)\frac{c-1}{c}n}{\lambda\gamma c\tau} \\
&\leq \frac{n}{c\tau} + \frac{\max_i \lambda_{\max}(A_i^\top A_i)}{\lambda\gamma}.
\end{aligned}$$

Therefore, the dominant term in (44) is a strict lower bound of that in (47). Moreover, it is clear that the gap between (44) and (47) is large when the data is sparse. For instance, in the perfectly sparse case with $\hat{\omega} = 1$, the bound (46) for Quartz becomes

$$\frac{n}{c\tau} + \left(1 + \frac{1}{\tau - 1}\right) \frac{\max_i \lambda_{\max}(A_i^\top A_i)}{\lambda \gamma c \tau},$$

which is much better than (47).