[Reviews · NeurIPS 2015]

Submitted by Assigned_Reviewer_1

The paper presents a variant of dual coordinate ascent for regularized empirical risk minimization with smooth convex loss functions and strongly convex regularizers. The authors derive an update scheme where in each iteration only a subset of dual coefficients are updated. The authors proof the convergence of their method for any sampling scheme producing those subsets. Moreover, they derive sampling schemes to speedup the convergence. In experiments, the authors verify their theoretical findings.

The paper is well-written and scientific-sounding. The theoretical as well as the experimental results suggest that the proposed method outperforms existing algorithms when trained highly parallel, especially for very sparse data. My only concern is about the experiments which are not very conclusive: - combination of dataset, setting and method is not consistent

- why not performing serial and mini-batch experiments for all 6 datasets?

- why not reporting SPDC-roh=1 together with the other methods the serial setting?

- likewise, why not reporting on ASDCA in the mini-batch setting? - no reporting of wall-clock time; and no reporting of predictive performance and/or values of the objective; it's hard to tell wether primal dual gaps of 1e-4 and 1e-10 make any difference in practise - no setting other than block size m (=1), loss (smoothed hinge), and regularizer (l2); consider m>1 where sparsity may significantly drop, Ridge regression where the Lipschitz constant is unbounded, etc. - only one large real-world dataset; consider rcv1 and/or imagenet - please detail on the choice of lambda. It has a significant impact on the convergence speed, but also on the predictive performance of the resulting model. For very large datasets, "optimal" lambda effectively approaches zero; which (practical) impact would this have on the convergence?

Minor comments for the authors: - line 224: consider "the number of matrices A_i with any nonzero entry in row j" (it wasn't quit clear to me what "blocks" refers to) - Eq 17: colon - line 295: should be omega_j; consider adding max_j omega_j \leq n - "average sparsity", in line 295, is misleading; omega_j tilde is the "average number of non-zero entries for dimension j"; omega is an (absolute) measure of density rather than a (relative) measure of sparsity. Similarly, in Table 3, where 100% sparsity denotes a fully dense dataset. - titles in Figure 1 and, in particular, in Figure 2 are not very helpful; consider adding the 3 artificial datasets in Table 3; currently the sparsity (their key property) is only stated in the caption of Fig 2. - Figure 1: consider reordering the methods where both Quartz variants should be red - supp.: last line before Ch. 8 should be m > 1
Summary: The authors propose a variant of dual coordinate ascent which is very efficient for very sparse data and can leverage distributed/parallel computing. The proposed method and the theoretical findings contribute to the fields, however, the experiments are not very detailed.

Submitted by Assigned_Reviewer_2

Quality ------- The paper is of high quality and the analysis appears technically sound and

complete. The empirical evaluation is rather brief, but shows that the theoretical

analysis matches the empirical performance.

Clarity ------- The paper is well written and structured, and is informative even without consulting the proofs in the appendix.

Originality ----------- While the proposed algorithm is very close to (variants of) SDCA, the extension to arbitrary sampling distributions and the corresponding analysis appear novel.

Significance ------------ Stochastic convex optimization is clearly a relevant topic, and the paper makes contributions both on the algorithmic as well as on the analysis side that other researchers can build on. For practitioners, the impact of the paper could be much improved by demonstrating the performance of the algorithm in the distributed setting, with data sets that do not comfortably sit in the memory of a single machine.
Summary: The paper proposes and analyzes a novel primal-dual algorithm for stochastic convex optimization. The paper is of high quality, well written (though dense) and the proposed algorithm and analysis appear novel. The (limited) empirical analysis shows some improvement over state of the art in certain regimes.

Author Feedback
Author rebuttal: Thanks to all reviewers for their comments.

Rev 1:

1) All your suggestions for additional tests are completely reasonable - in fact, we have done vastly more testing than what we have been able to include in the due to the page limit. We will do our best to try to add several new experiments to the main paper or suppl material.

2) We will add SPDC (\tau=1) to the serial setting plots and ASDCA to the mini-batch setting plots. We did test with large real-world datasets, such as astro_ph and CCAT. The performance in the serial setting is quite similar to what we report in the paper.

3) In the mini-batch setting, in our experience Quartz is better than SPDC whenever the "average number of nonzeros \tilde w" as defined in line 294, is sufficiently small. Notice that \tilde w in general does not only depend on the sparsity of the data. To demonstrate the performance of Quartz, we generated a random dataset with sufficiently small \tilde.

4) The computational costs per epoch of the considered methods (Quartz, SDCA, SPDC) are all of the same order as nnz(A). From this point of view, comparing the number of epochs is preferable to wall-clock time as it is independent of implementation and compute environment.

5) In the serial setting, lambda is chosen to be close to 1/n, which is a popular setting believed to lead to a good predictive performance. With lambda smaller than 1/n, the method will converge more slowly both in theory and in practice and in this case accelerated methods such as SPDC and ASDCA are normally a better choice. However, as shown by Fig 2, for certain datasets, even when lambda=0.1/n, Quartz can be better than the accelerated method SPDC. In general it is difficult to find a good balance between the convergence speed of the method (the choice of lambda) and a good predictive performance, the usual approximation-estimation trade-off.

6) We will fix all minor issues.

Rev 2:

Note that in Fig 2 we **do** show Quartz can be much better than SPCD. As shown in our experiments, for sufficiently sparse datasets, Quartz enjoys almost linear speedup (1/\tau) in the mini-batch setting but SPDC only has sublinear speedup (1/\sqrt(\tau)) where tau is the size of the mini-batch. This is an enormous difference for large mini-batch sizes!

Rev 3:

We will do our best to try to squeeze in an additional experiment - in a distributed setting.

Rev 5:

Thanks!

Rev 4 and 7:

Both reviewers question the benefit of our method (Quartz) in comparison with SDCA / I-prox SDCA since in their words the methods have the same complexity rate (we mention this in Sec 3.1) and similar empirical behavior (we mention this in Fig 1). In fact, the answer to your question is clearly articulated in many parts of the text, including the abstract, introduction, Sec 3.2, Sec 3.3 and Fig 2. Please read these!

1) The theoretical and practical performance of Quartz specialized to the serial sampling is indeed very close to SDCA/Iprox-SDCA. We considered it important to mention this for the benefit of the reader as this is an important special case. This is an observation meant to re-assure the reader already familiar with these state of the art algorithms - it is not the main selling point of the paper!

2) Note that while SDCA / Iprox-SDCA only update one dual coordinate at a time, Quartz allows for "arbitrary sampling" (a distribution over all subsets of coordinates) - which gives it massive flexibility and in our opinion will lead to many applications in the future. Quartz is the first primal-dual method capable of working with an arbitrary sampling.

3) When more than one processor is available, updating in parallel more than one dual coordinate can lead to significant speedup, provided that a suitable stepsize is carefully chosen to guarantee the convergence. Such mini-batch setting was not considered previously for SDCA nor Iprox-SDCA! This arises as a special case of our arbitrary sampling setup.

4) Quartz is capable of taking advantage of the sparsity of the data so that almost linear speedup (processing time divided by the number of processors used) is achieved when the data is sufficiently sparse. Such data-driven speedup property as demonstrated in Figure 2, does not exist in other SDCA-like methods; note that our method is vastly better than SPDC in such a regime.

5) The data driven speedup can be so large that our method can be better, both in theory and in practice (see Sec 3.3), than methods with momentum.

6) Our method is not merely a variant of SDCA, it is an entirely new method. Its analysis is simpler than that of SDCA, and unlike SDCA, it is directly primal-dual. As a result, in the special case of a serial sampling, we obtain the right logarithmic terms (this is not very important, but is nice to have nevertheless).